# Decoupling of mRNA and Protein Expression in Aging Brains Reveals the Age-Dependent Adaptation of Specific Gene Subsets

**DOI:** 10.3390/cells12040615

**Published:** 2023-02-14

**Authors:** Inès Khatir, Marie A. Brunet, Anna Meller, Florent Amiot, Tushar Patel, Xavier Lapointe, Jessica Avila Lopez, Noé Guilloy, Anne Castonguay, Mohammed Amir Husain, Joannie St. Germain, François-Michel Boisvert, Mélanie Plourde, Xavier Roucou, Benoit Laurent

**Affiliations:** 1Research Center on Aging, Centre Intégré Universitaire de Santé et Services Sociaux de l’Estrie-Centre Hospitalier Universitaire de Sherbrooke, Sherbrooke, QC J1H 4C4, Canada; 2Department of Biochemistry and Functional Genomics, Faculty of Medicine and Health Sciences, Université de Sherbrooke, Sherbrooke, QC J1H 5N4, Canada; 3Department of Pediatrics, Medical Genetics Service, Université de Sherbrooke, Sherbrooke, QC J1H 5N4, Canada; 4Centre de Recherche du Centre Hospitalier Universitaire de Sherbrooke (CRCHUS), Sherbrooke, QC J1H 5N4, Canada; 5Department of Immunology and Cell Biology, Faculty of Medicine and Health Sciences, Université de Sherbrooke, Québec, QC J1H 5N4, Canada; 6Department of Medicine, Faculty of Medicine and Health Sciences, Université de Sherbrooke, Sherbrooke, QC J1H 5N4, Canada; 7Quebec Network for Research on Protein Function, Structure, and Engineering, PROTEO, Québec, QC G1V 0A6, Canada

**Keywords:** aging, brain, cortex, mRNA, transcriptome, protein, proteome

## Abstract

During aging, changes in gene expression are associated with a decline in physical and cognitive abilities. Here, we investigate the connection between changes in mRNA and protein expression in the brain by comparing the transcriptome and proteome of the mouse cortex during aging. Our transcriptomic analysis revealed that aging mainly triggers gene activation in the cortex. We showed that an increase in mRNA expression correlates with protein expression, specifically in the anterior cingulate cortex, where we also observed an increase in cortical thickness during aging. Genes exhibiting an aging-dependent increase of mRNA and protein levels are involved in sensory perception and immune functions. Our proteomic analysis also identified changes in protein abundance in the aging cortex and highlighted a subset of proteins that were differentially enriched but exhibited stable mRNA levels during aging, implying the contribution of aging-related post- transcriptional and post-translational mechanisms. These specific genes were associated with general biological processes such as translation, ribosome assembly and protein degradation, and also important brain functions related to neuroplasticity. By decoupling mRNA and protein expression, we have thus characterized distinct subsets of genes that differentially adjust to cellular aging in the cerebral cortex.

## 1. Introduction

Aging is a gradual process of natural changes leading to the progressive loss of physiological integrity and the impairment of essential biological functions. This deterioration of physiological functions with advancing age is the primary risk factor for developing chronic diseases and increasing vulnerability to death. The cellular and tissue changes are sustained by molecular aberrations at different regulatory layers: DNA and chromatin, RNA and proteins. During aging, DNA integrity is challenged by genomic instability and telomere attrition. Protective and repair mechanisms fail to address the damages inflicted on DNA, and genetic damages tend to accumulate [1], leading to deleterious cellular effects such as senescence or apoptosis [2]. Age-related changes in gene expression are linked to epigenetic alterations. During aging, specific epigenetic alterations, such as the increase in H4K16ac, H4K20me3 and H3K4me3 histone marks (activation), the decrease in H3K9me and H3K27me3 histone marks (repression), or the global increase in DNA methylation level (repression), can directly impact the expression of the genes [3,4,5,6]. Finally, the ability to preserve protein homeostasis, also called proteostasis, under normal and stress conditions is gradually compromised with age [7,8]. These primary hallmarks of cellular aging are conserved across evolution and can lead to the loss of important cellular functions [9]. During aging, the overall mRNA translation rate decreases as the translation machinery becomes less efficient. This decrease is due to the deregulation of the stoichiometry of translation components, leading to the decreased abundance and activity of ribosomes [10,11] and the possible sequestration of mRNAs in stress granules, where they cannot be translated [12]. Therefore, the mRNA/protein expression ratio for a specific gene might be strongly affected during aging.

In a cell, the average transcription rate is about two mRNA molecules per hour, with a range between 0.1 and 100 mRNA molecules per hour depending on the given locus [13]. The median translation rate is about 140 proteins per mRNA per hour, with a maximum rate of 1000 proteins per mRNA per hour [13,14,15]. Moreover, proteins are, on average, 5 times more stable than mRNAs [13]. Therefore, it is easier to change the abundance of mRNAs to have an impact at the protein level. The mRNA level is mostly important for newly synthesized proteins since their initial concentration is minimal, whereas changes at the protein level are needed for pre-existing proteins with basic cellular functions [16]. Even though a higher concentration of mRNA molecules has more chances to be transcribed into many more proteins, there are several regulatory mechanisms in place modulating the level of protein expression: the translation rate, the rate of degradation, and also protein synthesis delay and transport [14]. The translation rate is primarily modulated by the mRNA sequence itself—codon composition, upstream open reading frames (uORFs), and the presence of internal ribosome entry sites (IRES) [17,18]. The translation rate can also be modulated by the availability of the ribosomes and the binding of different molecules to the mRNA, such as micro-RNA or long noncoding RNA (lncRNA). Therefore, two groups of genes exhibiting different types of regulation have been described: (i) genes that mainly rely on transcriptional regulation, with a change in mRNA level correlating with that of proteins, and (ii) genes that depend on a translational regulation, where the RNA dynamic is not reflected into protein levels because of post-transcriptional mechanisms [19,20]. In a steady state, mRNA levels primarily explain protein levels as their respective average remains stable over time even when cells undergo long-term dynamic processes such as proliferation or differentiation [21,22,23]. For short-term adaptation, post-transcriptional processes are more important than the initial levels of mRNA as the regulation of transcript levels alone would be too slow. An increase in translation for already existing transcripts or an increase in protein degradation is a faster and more efficient mechanism. There is a poor mRNA–protein correlation during stress as it causes a significant change in translation rate and protein misfolding [24,25]. During aging, mRNA and protein expression levels can drastically change [12,26,27,28]. The age-dependent association between mRNA and protein levels remains mainly unexplored, and it is not known whether cellular aging impacts specific groups of genes.

In mice, transcriptomic studies on different tissues with bulk RNA-seq experiments have shown that most genes age with the same pattern in different tissues but differ in amplitude and onset of expression depending on the organ [29,30]. The brain is one of the organs that age the fastest [31]. Age is the primary risk factor for cognitive impairment as well as for the development of neurodegenerative diseases such as Alzheimer’s disease. Here, we have investigated the connection between changes in mRNA and protein expression in the aging brain by comparing the transcriptome and proteome of the cerebral cortex during normal aging. Our transcriptomic analysis showed that aging mainly triggers gene activation in the cortex and that genes with increased mRNA and protein expression are associated with sensory perception and immune functions. Our proteomic analysis revealed changes in protein abundance in the aging cortex and highlighted a subset of proteins that were differentially enriched but exhibited stable mRNA levels during aging. These genes, targeted by aging-dependent post-transcriptional mechanisms, are associated with general biological processes (i.e., translation, ribosome assembly, protein degradation) but also have functions related to neuroplasticity (i.e., neurogenesis, dendrite outgrowth, synapse formation). By decoupling mRNA and protein expression, we have characterized distinct subsets of genes that differentially adjust to cellular aging in the cerebral cortex.

## 2. Materials and Methods

### 2.1. Mice

Male C57BL/6J mice were purchased from the Réseau Québécois de Recherche sur le Vieillissement (RQRV, Québec City, QC, Canada). All mice received a standard diet, i.e., NIH-31 modified open formula mouse/rat sterilized diet (Teklad 2019, Harlan Laboratories), and were kept on a 12:12 daylight cycle at 22–23 °C with access to a small house and tissues for nesting. Adequate measures were taken to minimize pain and discomfort. Animal care and handling were performed in accordance with the Animal Care Committee of Université Laval (CPAUL), the guidelines of the Canadian Council on Animal Care and the Université Laval institutional policy.

The weight of each mouse was taken before euthanasia. Mice were placed under anesthesia (isoflurane), and then a cervical dislocation was performed. The brain was first removed from the cranial cavity, washed in phosphate-buffer-saline (PBS) and then weighed before the microdissection of the cerebral cortex. If mice presented visible tumors in any major organs (for example, the liver), they were discarded from the study.

### 2.2. Histology

Whole brains from 6- and 24-month-old mice were put in a plastic cassette, submerged in paraformaldehyde (PFA) 4% for 16 h at 4 °C. Then, several washes of fresh ethanol 70% were performed for 3 × 5 min, followed by incubating in ethanol 70% for 12 h at 4 °C. Fixed brains were sent to the Histology platform of the Université de Sherbrooke (Sherbrooke, QC, Canada) to be embedded in paraffin and cut in a 4 µm coronal cross-section (around Bregma 0). The cortical thickness and the surface area of the cortical layer were measured on brain slides stained with hematoxylin and eosin. The measurement was blind for the age group and performed on 2 slides per brain for 6 mice per age group. 

### 2.3. RNA and Protein Extraction

Each frozen sample of cortex was crushed and powdered using the BioSqueezer Snap Freeze Press system (BioSpec Products, Bartlesville, OK, USA), to allow homogenization and equal fractionation for both transcriptomic and proteomic studies. The equipment was previously cooled using dry ice. The subsequent powder was then fractionated and stored at −80 °C in multiple Eppendorf tubes for RNA or protein extraction.

To isolate total RNA from the tissue powder, a hybrid technique combining the TRIzol reagent protocol (Invitrogen, Thermo Fisher Scientific, Waltham, MA, USA) and the Quick-RNA™ Miniprep Kit (Zymo Research, Thomas Scientific, Swedesboro, NJ, USA) was used. Cortex powder was first homogenized in 500 µL of cold TRIzol, and then 500 µL of chloroform:isoamyl alcohol 24:1 (Sigma Aldrich, St. Louis, MO, USA) was added. Samples were vortexed for 20 s and incubated for 2 min at room temperature. The samples were then centrifuged at 10,000× *g* for 18 min at 4 °C. The aqueous phase, which contains RNAs, was transferred to a new Eppendorf tube. An equal volume of 100% ethanol was added, and the mixture was then loaded into a Zymo-Spin™ IIICG Column from the Quick-RNA™ Miniprep Kit. Further steps were conducted according to the manufacturer’s protocol. The RNA concentration in ng/µL was determined using a Nanodrop system (Thermo Fisher Scientific).

To isolate proteins, tissue powder was resuspended in white Laemmli 1× (62.5 mM TRIS pH 6.8, 10% v:v glycerol, 2% v:v SDS) and boiled for 5 min at 95 °C. Samples were then sonicated by bursts of 5 sec at 12% amplitude, then 10 sec on ice, for a total of 5 times. Absolute protein quantification was performed by bicinchoninic acid assay (BCA) using the Pierce™ BCA Protein Assay Kit according to the manufacturer’s parameters. The absorbance was read at 562 nm using a spectrophotometer, and the absolute quantification was calculated.

### 2.4. RNA-Seq Library Preparation

RNA-seq libraries were prepared using NEBNext^®^ Single Cell/Low Input RNA Library Preparation Kit (New England Biolabs, Ipswich, MA, USA). Library preparation was initiated with 10 ng of total RNA. The RNA, along with all the other reagents, was thawed on ice. Further steps were conducted according to the manufacturer’s protocol until the final library amplification. Each final library was then purified using the MinElute PCR Purification Kit (Qiagen, Hilden, Germany) and then quantified on a Qubit system (Thermo Fisher Scientific). Multiplexed libraries were pooled in approximately equimolar ratios and were then run in a 2% agarose gel. The DNA between 300 and 700 base pairs was then cut and purified using the Qiagen Gel Extraction Kit (Qiagen). Library sequencing was performed at the RNomics platform of the Université de Sherbrooke (Sherbrooke, QC, Canada). Single-end sequencing was performed on all the samples with a 75 bp sequencing depth on a NextSeq machine. The raw data have been deposited at the Gene Expression Omnibus (GEO) under the subseries entry GSE218637.

### 2.5. Data Analysis

Quality control of RNA-seq data was performed using FastQC (version 0.11.8) with default parameters. Trimming of the reads was achieved using TrimGalore (version 0.6.4) with default parameters except for: --illumina --max_n 5 --clip_R1 3 --three_prime_clip_R1 3. The quality of the data was also evaluated using FastQC after the trimming of adapters and low-quality regions. Reads were then mapped to the mouse genome (GRCm38, GENCODE vM23, primary assembly) with STAR (version 2.7.3a). STAR was used with default parameters except for: --outFilterMultimapMmax 10 --outSAMtype BAM SortedByCoordinate --quantMode GeneCounts --outSAMprimaryFlag AllBestScore --outFilterMismatchNmax 5. The evaluation of gene expression (CPM and TPM) was done using FeatureCounts from the SubRead package (version 2.0.0). FeatureCounts was used with default parameters except for: -M --fraction -Q 15 --fracOverlap 0.25 --largestOverlap --minOverlap 15. Exons were set as features (-t exon) and grouped by gene (-g gene_id). Statistical analysis was performed using the edgeR package (version 3.24.3) from R (version 3.5.3), and significant changes in gene expression (1% FDR) were evaluated after filtration using the genefilter package (version 3.15), as previously done [32]. Data visualization was done using python (version 3.7.3).

### 2.6. Quantitative PCR 

Total RNA (500 ng) was used to generate cDNA using the Biorad iscript RT supermix RT-qPCR kit (Biorad, Hercules, CA, USA) following the manufacturer’s instructions. The expression of different target genes was validated by quantitative PCR (qPCR) using the Azur Cielo 3™ system (Azure Biosystems, Dublin, CA, USA). The specific primers used for the amplification are listed below:

RPLP0 F: CTGAGATTCGGGATATGCTGTTGGCC

RPLP0 R: CGGGTCCTAGACCAGTGTTCTGAG

CYP26B1 F: CCAATTCCATTGGCGACATCCACCG

CYP26B1 R: GGCAGGTAGCTCTCAAGTGCCTC

SERPINB9B F: CACGTGGGTCTCCAAACAGACTGAAG

SERPINB9B R: CATCATCTGCACTGGCCTTTTCTCATC

LILRB4 F: GACCTCATGATCTCAGAAACCAAGGAC

LILRB4 R: GTTCTCAGATTGTGTGTTCTTCACAGAAGC

KLF10 F: CTCCAGCAAGCTTCGGAGGGGAAA

KLF10 R: GGGGCTGTAAGGTGGCGTTAAACA

CCL8 F: CCTGTCAGCCCAGAGAAGCTGACTG

CCL8 R: CAGAGAGACATACCCTGCTTGGTCTGGA

GADD45a F: GCAGGATCCTTCCATTGTGATGAATGTGG

GADD45a R: CTAGCTGAGCTGCTGCTACTGGAGAAC

KCNG1 F: ATGAGGAGGCTGCGTGACATGGTGGA

KCNG1 R: GGATGAAGCGCAGCAGGAACTCTAG

Egr2 F: GTCTGGTTTCTAGGTGCAGAGATGGG

Egr2 R: CCTTTGACCAGATGAACGGAGTGGC

CXCL13 F: CAGGCAGAATGAGGCTCAGCACAG

CXCL13 R: CTTCATCTTGGTCCAGATCACAACTTCAG

BC1 F: CCTGGGTTCGGTCCTCAGTCTGGA

BC1 R: GGTTGTGTGTGCCAGTTACCTTG

GAP43 F: CCTCCAACGGAGACTGCAGAAAGC

GAP43 R: TCAGGCATGTTCTTGGTCAGCCTCG

The reactions were performed with the PerfeCta SYBR Green SuperMix (QuantaBio, Beverly, MA, USA), as recommended by the manufacturer. Real-time PCR was performed with a hot start step at 95 °C for 2 min, followed by 40 cycles of 95 °C for 10 s, 60 °C for 10 s and 72 °C for 20 s. Analysis was performed using Azur Cielo software (Azure Biosystems, Dublin, CA, USA). The relative expression of the genes was normalized by that of RPLP0. 

### 2.7. Sample Preparation for Mass Spectrometry

*Reduction, Alkylation of Proteins*. All buffers used in this stage were prepared with MS-grade water. The protein reduction step was carried out by incubating the protein in Laemmli with 10 mM DTT for 30 min at 60 °C. The alkylation of the proteins was carried out by adding 15 mM iodoacetamide (Sigma Aldrich, St. Louis, MO, USA) for 30 min at room temperature away from light. In order to increase the depth of discovery, the proteins were separated by size on SDS-PAGE gel; 20 µg of proteins were loaded onto 4–20% PROTEAN^®^ TGX™ precast protein gels (Biorad, Hercules, CA, USA). The gel was stained with Invitrogen™ SimplyBlue™ SafeStain (Thermo Fisher Scientific, Waltham, MA, USA) according to the manufacturer’s instructions. The gel was then cut into 4 fractions, ranging from 250–75, 75–37, 37–20, and 20–10 kDa. Each fraction was diced into 1 mm cubes, and then the gel pieces were destained. All incubation steps were performed at 30 °C at 800 RPM in a thermoshaker. The pieces were first washed in MS-grade H20 for 30 min. Then, we added the same volume of CH_3_CN for a final concentration of 50% v:v CH_3_CN:H_2_O, incubated for 15 min. The supernatant was removed, and the pieces were resuspended in 20 mM NH_4_CO_3_ for 15 min. Then, the supernatant was removed, and the pieces were resuspended in 50% v:v CH3CN and 10 mM NH_4_CO_3_ for 15 min. The last two steps were repeated until no trace of blue was visible in the gel pieces. The proteins were digested by adding 1 µg Pierce MS-grade trypsin (Thermo Fisher Scientific, Waltham, MA, USA) diluted in 20 mM NH_4_CO_3_ and incubated overnight at 37 °C.

*Peptide retrieval*. The following incubation steps were all performed at 30 °C at 850 RPM in a thermoshaker. An equal volume of CH_3_CN (equivalent to that of the trypsin NH_4_CO_3_) was added to the gel pieces and incubated for 30 min. The supernatant containing the peptides was transferred into low-bind Eppendorf™ microcentrifuge tubes. An equal volume of 1% (v:v) formic acid (FA) (Thermo Fisher Scientific) was added to the gel pieces, and the sample was incubated for 20 min. This second supernatant was transferred to the same tube as the first supernatant. The steps of adding FA and transferring the supernatant were repeated once. Then, 150 µL of CH_3_CN was added, and the mixture was incubated for 15 min. Again, the supernatant containing the last peptides is transferred to the tube. These samples were thereafter concentrated by a centrifugal evaporator at 65 °C until complete drying (4–6 h) and resuspended in 30 µL of 0.1% trifluoroacetic acid (TFA) buffer.

*Purification and Desalting of the Peptides on C18 Columns*. The peptides were purified with ZipTip 10-l micropipette tips containing a C18 column (EMD Millipore, Burlington, MA, USA). The ZipTip was first moistened by suctioning 10 µL of 100% acetonitrile (ACN) solution three times, then equilibrated by suctioning 10 µL of 0.1% TFA buffer three times. Each peptide sample was passed on the balanced ZipTip by ten up-and-downs of 10 µL of the sample. This step was performed three times to pass the entire sample to the column. The ZipTip was then washed with 10 µL of 0.1% TFA buffer three times. 

### 2.8. Data-Dependent Acquisition (DDA) Mass Spectrometry

*Mass spectrometry*. Trypsin-digested peptides were separated using a Dionex Ultimate 3000 Binary RSLCnano instrument for high-performance liquid chromatography (HPLC); 1.5 µg of peptides per fraction were loaded onto an OrbiTrap mass spectrometer. With a constant flow of 4 µL/min, the samples were loaded and separated onto a nanoHPLC system (Dionex Ultimate 3000, Dionex, Sunnyvale, CA, USA) with a trap column (Acclaim PepMap100 C18 nano column, 0.3 mm id × 5 mm, Dionex, Sunnyvale, CA, USA). Peptides were then eluted off towards an analytical column heated to 40 °C (PepMap C18 column (75 µm × 50 cm)), with a linear gradient of 5–45% of solvent B (80% ACN with 0.1% formic acid) over a 4 h gradient at a constant flow of 450 nL/min. Peptides were analyzed using an OrbiTrap QExactive instrument (Thermo Fischer Scientific) using an EasySpray source at a voltage of 2.0 kV and the temperature of the column set to 40 °C. Acquisition of the full scan MS survey spectra (*m/z* 350–1600) in profile mode was performed in the OrbiTrap at a resolution of 70,000 using 1,000,000 ions. The ten most intense peptide ions from the preview scan in the OrbiTrap were fragmented by collision-induced dissociation (normalized collision energy 35% and resolution of 17,500) after the accumulation of 50,000 ions with maximum filling times of 250 ms for the full scans and 60 ms for the MS/MS scans. All unassigned charge states, as well as species with single, seven or eight charges for precursor ions, were rejected. A dynamic exclusion list was set to 500 entries with a retention time of 40 s (10 ppm mass window). Data acquisition was performed using Xcalibur (version 4.2). The mass spectrometry proteomics data have been deposited to the ProteomeXchange Consortium via the PRIDE partner repository [33] with the dataset identifier PXD038179.

*Data analysis*. Data analysis was performed using MaxQuant software [34] (1.6.17). For statistical analysis, the Prostar software package [35] (Prostar 1.26.3; DAPAR 1.26.1) was used with the following settings. Potential contaminants, as well as the reverse sequences identified, were removed, and only proteins with at least two unique peptides detected in all replicates of at least one condition were kept. Partially observed values (POVs) were imputed using the SLSA (Structured Least Square Adaptative) regression-based imputation method, while values missing in the entire condition (MEC) were imputed using DetQuantile with a low deterministic value. Hypothesis testing was performed using the Limma test. The log2 fold change threshold was set at 1 and the –log10 p-value threshold at 1.3. Results were visualized on a volcano plot using the VolcaNoseR tool [36]. From the significantly different proteins, a heatmap was generated using the Heatmapper tool [37] with average linkage clustering and Pearson distance measurement. Peak list files were searched from the Openprot mouse proteome (openprot.org; version 1.6). 

### 2.9. Parallel Reaction Monitoring (PRM) Mass Spectrometry

Protein quantification was performed using a label-free PRM method. For quantification on an OrbiTrap mass spectrometer, the 3 fractions containing the peptides of interest were pooled in equal concentration and were loaded and separated onto a nanoHPLC system (Dionex Ultimate 3000) with a constant flow of 4 µL/min onto a trap column (Acclaim PepMap100 C18 nano column, 0.3 mm id × 5 mm, Dionex, Sunnyvale, CA, USA). Peptides were then eluted off towards an analytical column heated to 40 °C (PepMap C18 column; 75 µm × 50 cm), with a linear gradient of 5–45% of solvent B (80% ACN with 0.1% formic acid) over a 4-h gradient at a constant flow of 450 nL/min. Eluted peptides were analyzed on an OrbiTrap QExactive instrument (Thermo Fischer Scientific) using an EasySpray source at a voltage of 2.0 kV. Acquisition of the MS/MS spectra (*m*/*z* 350–1600) was performed in the OrbiTrap. An inclusion list containing the *m*/*z* values corresponding to the monoisotopic form of the selected peptides was generated. The collision energy was set at 28% at 140,000 for 1,000,000 ions, with maximum filling times of 250 ms and an insulation width of 0.6. Data acquisition was performed using Xcalibur (version 4.2). The mass spectrometry proteomics data have been deposited to the ProteomeXchange Consortium via the PRIDE partner repository [33] with the dataset identifier PXD038179.

The identification and quantification of peptides were performed on Skyline software (21.1.0.146). The relative amount of proteins was calculated using peptide peak areas extracted from retention time points for all fragment ions (if possible, the 3–5 most abundant ions were chosen for quantification for each peptide) and normalized against peptides of endogenous proteins whose abundance did not change in the samples. Normalized values were then used to calculate ratios between the two experimental groups (24-month/6-month), and values of technical replicates were used for data visualization.

### 2.10. Immunoblotting

The proteins were resuspended in 1× Laemmli (62.5 mM Tris pH 6.8, 25% *v*/*v* glycerol, 2% *m*/*v* SDS, 0.01% *m*/*v* bromophenol blue, 10% *v*/*v* 1M DDT) and then boiled for 5 min at 95 °C; 20 μg of each sample were then analyzed on 10% SDS-PAGE gel in 1× running buffer at 100 V for around 90 min. Proteins were transferred to a nitrocellulose membrane (Whatman) with 1× Tris-glycine buffer at 100 V for 1 h. The membranes were blocked with 5% *m*/*v* powdered milk dissolved in 1× PBS—0.1% Tween for 30 min and then washed 3 times for 5 min in 1× PBS—0.1% Tween with agitation at room temperature. Membranes were incubated overnight at 4 °C with a primary antibody. The next day, membranes were washed 3 times for 5 min in 1× PBS–Tween at room temperature and incubated for 1 h at room temperature with the horseradish peroxidase (HRP)-conjugated secondary antibody. After this incubation, the membranes were washed again 3 times for 5 min with 1× PBS—0.1% Tween. Results were visualized on an iBright machine (Thermo Fisher Scientific). The intensity of the GAP43 signal was quantified using iBright analysis software and normalized to that of the actin signal. The antibodies used for immunoblotting are listed in Table 1.

### 2.11. Immunofluorescence

Brain slides were first incubated for 20 min in Citrisolv (1×) for deparaffinization. For rehydration, brain slides were washed two times with EtOH 100% for 3 min and then sequentially washed with EtOH 95%, EtOH 80% and water for 1 min. Slides were then incubated into boiling sodium citrate (temperature around 95 °C) for 20 min and then cooled in the same solution for 20 min without heat. Slides were then washed 2 times with PBS 1x. For permeabilization and blocking, slides were incubated with a blocking solution (PBS 1× + 0,1% Triton + 4% BSA) at room temperature for 2 h minimum in a closed, humidified chamber. The primary antibody was then added to the brain sections and incubated overnight at room temperature. The next day, slides were washed 3 times for 10 min with PBS 1×, and a fresh secondary antibody solution (dilution 1:1000) was added for 45 min in the dark at room temperature. Slides were then washed 3 times with PBS 1× for 10 min each. A drop of ProLong Glass Antifade Mountant (DAPI, Thermo Fisher Scientific) was finally added to the center-left of the tissue with a coverslip on top. All brain slides were analyzed on the Axioscope 5/7/Vario system (Zeiss, Oberkochen, Germany) with the same acquisition settings. The acquisition and analysis of images were blind for the age group. The antibodies used for immunofluorescence are listed in Table 2.

## 3. Results

### 3.1. Cortical Thickness Increases during Aging

Throughout this study, two age groups of C57BL/6J mice were used to investigate cerebral cortex aging: young adults and old adults. Aging studies typically used 3-month-old and 18-month-old mice to respectively represent the two populations; however, important morphological modifications occur up to 4 months of age, such as myelination [38,39]. To ensure that molecular differences with older mice do not arise from developmental and physiochemical changes related to the developmental stages, we used 6-month-old mice to represent the age group of young adults. In the same way, 18 months of age in mice (equivalent to 55 years in humans) do not represent the older stages of life. Hence, we used 24-month-old mice (equivalent to 75 years in humans) for the age group of old adults because biomarkers of old age are prominently detected in animals of that age [40].

We first investigate the anatomical changes in the brain in young adult (6-month-old) and old adult (24-month-old) mice (n = 6 per age group; three males and three females). We first measured the cortical thickness on brain slides stained with hematoxylin and eosin (Appendix A). Since the cortical thickness varies in a coronal brain cross-section, the top layer being thicker than the side one, we measured the thickness of the “top” region (in blue in Figure 1A) and the “bottom” region (orange in Figure 1A), with measurements made in several locations, as indicated in Figure 1A (respectively 4 and 5 measurements for the top and bottom regions). We observed that the top cortical layer, corresponding to the anterior cingulate, motor and primary somatosensory areas of the cortex, was significantly thicker in 24-month-old brains (1.41 ± 0.25 mm) compared to 6-month-old brains (1.19 ± 0.18 mm), while we did not find any significant difference for the bottom region (0.75 ± 0.23 mm vs. 0.65 ± 0.24 mm) (Figure 1B). We also measured the total surface area of the cortical layer and showed that the overall cortex area was not significantly different between the young and old adult groups (respectively 9.37 ± 0.94 mm^2^ vs. 10.28 ± 1.09 mm^2^) (Appendix A). We observed a significant age-dependent increase in the body weight for females (22.87 ± 1.37 g vs. 26.73 ± 3.24 g) but not for the males (34.65 ± 3.73 g vs. 33.9 ± 4.44 g) (Appendix A). However, we showed that males and females exhibited the same wet brain weight throughout their lifespans (489 ± 13 mg at 6 months of age vs. 490 ± 19 mg at 24 months of age) (Figure 1C). It indicated that the increase in cortical thickness was not associated with an age-related change in wet brain weight. 

### 3.2. Aging Mainly Triggers Activation of Gene Expression in the Cortex

We next investigated changes in mRNA expression in the aging cortex. We microdissected the cerebral cortex of 6- and 24-month-old male mice (n = 3 for each age group), extracted total RNAs from the tissue and then performed a transcriptomic analysis (Appendix A). First, the principal component analysis (PCA) showed that the global gene expression patterns of 6-month-old cortices were very similar (Figure 2A). The global gene expression pattern of 24-month-old cortices was distinct from that of 6-month-old cortices, as suggested by the average position of the samples (cross) on the *x*-axis, which represents the first principal component responsible for 80% of the variance (Figure 2A). We also noted that the gene expression within the 24-month-old group was more variable, as suggested by the position of each sample regarding the second principal component, which was only responsible for 5% of the total variance. Together, these results indicated that 6- and 24-month-old cortices exhibited a distinct global gene expression signature.

We next analyzed each individual gene and their differential expression between 6 and 24 months of age and represented them in a volcano plot (Figure 2B). Our analysis showed that 1586 genes are differentially expressed, with a false discovery rate (FDR) at 1% and a p-value inferior to 10^−3^. Among those differentially expressed genes (DEGs), the majority were up-regulated during aging, sometimes up to 7 times, while few DEGs were down-regulated with a minor reduction of their expression (Figure 2B). We next selected DEGs with a stringent FDR of 0.1%. The expression pattern of these genes at 6 and 24 months is represented in a heatmap (Figure 2C). With these parameters, we show that 427 genes were differentially expressed in the cortex between 6 and 24 months of age. As shown in Figure 2B, the majority of these DEGs were up-regulated (n = 400), while only 27 genes were down-regulated (Figure 2C; Appendix A). Interestingly, we found that genes whose expression was up-regulated during aging were significantly overrepresented in a 6 Mb window of several chromosomes (chromosomes 2, 5, 9, 10, 17, 19) compared to the density of genes in the background (Appendix A). We thus identified eight clusters of genes that were statistically enriched in specific regions of chromosomes. We did not observe a similar distribution for the down-regulated genes (Appendix A). 

We finally performed a gene ontology (GO) analysis to identify which specific pathways or functions those genes were involved in. Using ShinyGO, a tool for in-depth analysis of gene sets [41], we showed that up-regulated genes (n = 400) were mainly involved in the biological process of sensory perception, while the 27 down-regulated genes were mostly involved in neuron-associated functions such as neuron development and differentiation or metal ion transport (Figure 2D–E). Gene-set enrichment analyses with the BioPlanet database, which catalogs 1658 pathways with their healthy- and disease-state annotations, also revealed multiple pathways enriched for the up- and down-regulated genes. We found that genes whose expression was increased in the cortex during aging were mainly involved in the cell surface interaction or adhesion of multiple immune cells (i.e., granulocytes, monocytes) (Figure 2F). Many up-regulated genes encode proteins implicated in the immune response, such as cell adhesion molecules (Cd244a, Cd19), receptors (Cd207, Ccr4, Lilbr4) or inflammatory chemokines (Ccl8, Cxcl13). A similar analysis for the down-regulated genes revealed that they were mostly involved in the BDNF signaling pathway, transmission across synapses or glutamate receptor trafficking, all pathways mainly linked to neuron functions (Figure 2G). Together, these results show that in the cortex, aging mainly triggers the expression of genes related to sensory perception and immune response.

### 3.3. Increased mRNA Expression Correlates with Higher Levels of Proteins

To validate our transcriptomic results, we then performed quantitative PCR (qPCR) on more cortex samples (n = 8 per age group, four males and four females). Among the 1586 DEGs with a 1% FDR (Figure 2B; Appendix A), we selected candidates with an overall good mRNA expression and/or with biological relevance to the functions of interest, e.g., immune response. We picked five down-regulated genes (*Klf10*, *GADD45a*, *KCNG1*, *Cyp26b1*, *Egr2*) and five up-regulated genes (*CXCL13*, *BC1*, *Serpinb9b, Lilrb4* and *Ccl8*). Our qPCR results confirmed the differential expression of all genes in the aging cortex with, for instance, the expression of Cyp26b1 decreased by 57% (1.00 ± 0.35 at 6 months vs. 0.43 ± 0.26 at 24 months) or the expression of Lilrb4 increased by 155% (1.00 ± 0.40 at 6 months vs. 2.25 ± 0.90 at 24 months) (Figure 3A). Even though higher levels of mRNA have more chances to produce more proteins, age-dependent regulatory mechanisms can modulate the level of protein expression. We then investigated whether changes in the mRNA level correlated with those of proteins. We first performed mass spectrometry (MS) analysis on the cortex samples of 6- and 24-month-old male mice (n = 2 for each age group) using a bottom-up MS technique, also known as “shotgun” in the Data-Dependent-Acquisition (DDA) mode. To compare more precisely the mRNA and protein levels, we used proteins extracted from the exact same cortex samples employed for the transcriptomic study. For 6- and 24-month-old cortices, we respectively extracted 101.9 ± 23.5 μg of proteins per mg of tissue and 119 ± 30.9 μg of proteins per mg of tissue (Appendix A), suggesting that the total protein quantity remains quite stable in the aging cortex. Before the initial filtering steps (i.e., removal of contaminants and reverse sequences, maintenance of proteins with at least two unique peptides in all replicates), we identified a total of 2471 proteins in both young and old adult cortices (Appendix A). We overlapped DEGs identified by RNA-seq (n = 427) with the list of proteins identified by DDA MS, but unfortunately, few DEGs (n = 13) were detected at the protein level by MS (Appendix A). We showed that proteins detected by MS mainly originated from the most expressed transcripts (Appendix A) and that transcripts found differentially expressed in the aging cortex are not among the most expressed in general (Appendix A; Appendix A). Among these 13 DEGs detected by MS, the majority (10/13) were down-regulated during aging and involved in neuronal functions, as shown in Figure 2E. We previously showed that genes up-regulated in the aging cortex were mainly coding for proteins implicated in the immune response (Figure 2F), and since the immune cell population is in the minority in the cerebral cortex, it might, hence, be difficult to detect these proteins by DDA MS, as further discussed. To override this technical limitation and confirm that an increase in mRNA expression leads to more protein expression in the aging cortex, we next performed immunofluorescence on 6- and 24-month-old brain slides. Among our candidate genes, the *Ccl8* gene was selected because its mRNA expression fold change was one of the highest (Figure 3A), and Ccl8 antibodies were commercially available and already tested. Ccl8 immunofluorescence on coronal cuts of 6-month-old brains showed cytoplasmic staining for cells mainly located in the motor area (MO) of the cortex (Figure 3B,C; top panel). Interestingly, the anterior cingulate area (ACA) of the cortex did not exhibit any significant Ccl8 immunostaining at 6 months of age, but cells in this specific cortex area displayed a strong increase in Ccl8 signals at 24 months of age (Figure 3C,D). We did not observe any significant increase in Ccl8 signals in the motor area of the cortex during aging (Figure 3C). Together, these results show that in the aged cortex, increased Ccl8 mRNA expression correlates with a higher level of proteins, specifically in the ACA.

### 3.4. A Subset of Genes Depends on Translation Regulation in the Aging Cortex

With the results of DDA MS on 6- and 24-month-old cortices, we next investigated whether some genes exhibited differential abundance in protein levels during aging. The distribution of proteins showing significant differential enrichment in the aging cortex was represented by a volcano plot and a heatmap (Figure 4A–B). The volcano plot displayed an extreme distribution for some proteins that were originally not identified in one age group (Figure 4A; Appendix A). We identified 164 proteins enriched in the old adult cortex and 42 proteins specifically enriched in the young adult cortex (Figure 4A; Appendix A). Using ShinyGO, we showed that proteins up-regulated in the aging cortex were mainly involved in multiple catabolic and metabolic processes (including the proteasome complex), fatty acid beta-oxidation, organelle organization or mRNA nonsense-mediated decay (Figure 4C). Interestingly, we found significant enrichment for members of the PSMD and PSMC families (PSMD3, PSMD5, PSMD6, PSMD11, PSMD13, PSMC1, PSMC4, PSMC5), both encoding proteasome 26S subunits involved in the degradation of damaged, misfolded, abnormal and foreign proteins [42,43] (Appendix A). A similar analysis for proteins down-regulated during aging revealed that they were mostly involved in translation, mitochondrial membrane organization or ribosome assembly (Figure 4D). For instance, we found significant enrichment for ribosomal proteins (RPL18, RPL19, RPL23a, RPL38, RPL7a, RPL8, RPL9, RPS10, RPS26) (Appendix A). We next examined whether genes encoding proteins with significant differential enrichment during aging exhibited changes in mRNA expression. We showed that genes encoding differentially enriched proteins during aging had a stable mRNA expression in the cortex between 6 and 24 months of age (Figure 4E), suggesting that this subset of genes was the target of post-transcriptional mechanisms during aging.

To confirm our findings, we carried out a precise quantification of protein abundance using parallel reaction monitoring (PRM) MS (Appendix A). This technique requires us to select beforehand the protein and unique peptide(s) of interest to be targeted by MS. The technical limitations of the Q Exactive mass spectrometer allow the quantification of approximately 20 proteins with two quantotypic peptides per protein (i.e., unique peptides that reflect the stoichiometry of the protein). We selected 42 unique peptides from 22 different proteins, including two housekeeping proteins for normalization (Atp5b, Got 2) (Appendix A). For each sample, the same peptide solution used for the DDA method was re-injected into the mass spectrometer, with the list of the m/z values for these 42 peptides to allow their systematic detection (Appendix A). We analyzed the enrichment of each peptide in 6- and 24-month-old cortex samples compared to control peptides and have represented the results as a heatmap of the ratio of these peptides between each age group (Figure 4F). We confirmed with the PRM method that most selected peptides exhibited a change in intensity ratio, with some peptides showing a strong increase (Aldh2, Aldoc, Ctbp1, Mapk1, Psmd11, Tomm70a) and others a strong decrease (Apoa1, Apoe, App, Atp2b2, Cadm2, Cntn1, Ndufb10, Phf2, Prkcb, Prrt2) (Figure 4F). Another way to represent the results is to graph the area relative to the internal standard. When represented by a graph rather than a heatmap, we were able to confirm a decrease in Apoe, Atp2b2, Apoa1, and Prrt2 peptides and an increase in the Syn2 peptide (Figure 4G). Together, our findings show that, in the aging cortex, a subset of genes is regulated at the post-transcriptional level and that the mRNA dynamic is not reflected in the protein levels.

### 3.5. Aging Increases Protein Expression of Genes Associated with Neuroplasticity

Among the 165 proteins enriched in the old mouse cortex, we observed that many proteins also played a significant role in neurogenesis, axonal guidance, dendrite outgrowth and synapse formation, such as Camk2d, Cacnb3, Dlgap1, Dpysl5, Gabra1 or Gap-43 [44,45,46,47,48,49] (Appendix A). Evidence suggested that older brains had substantial plasticity, as illustrated by patients with stroke showing dramatic recovery with sustained therapy [50]. We next investigated the age-related changes in neuroplasticity, considering Gap-43 (growth associated protein 43) as a marker of brain-adaptive capabilities. Indeed, Gap-43 is expressed in axonal growth cones during development and regeneration [51]. We first examined Gap-43 expression by immunoblotting on 6- and 24-month-old cortex samples and observed that Gap-43 protein was indeed significantly increased during aging (1 ± 0.07 at 6 months vs. 2.17 ± 0.14 at 24 months) (Figure 5A; Appendix A; Table 1). We confirmed by qPCR that the mRNA expression level was stable between 6 and 24 months of age (1 ± 0.01 at 6 months vs. 0.96 ± 0.04 at 24 months) (Figure 5B). We finally performed immunofluorescence on 6- and 24-month-old brain tissue. Gap-43 immunostaining on coronal cuts of 6-month-old brains showed a good and specific signal in the motor and anterior cingulate cortices (Figure 5C, top panel; Appendix A). We observed a strong and significant increase of signal in the same brain areas at 24 months of age (4285 ± 944 AU at 6 months vs. 7181 ± 593 AU at 24 months) (Figure 5C, bottom panel). The results confirmed that Gap-43 expression increased at the protein level in the aging cortex, while its mRNA expression level remained stable, suggesting that post-transcriptional regulatory mechanisms are involved during aging.

## 4. Discussion

As we age, the volume of the brain reduces at the rate of around 5% per decade, starting at the age of 40 [52]. In middle-aged adults, wide-spread cortical thinning has been observed, reflecting significant global atrophy [53]. However, older age is associated with greater cortical thickness in the dorsolateral and anterior cingulate cortices [54]. Our findings support this observation as we have described an increase in cortical thickness between 6 and 24 months of age (Figure 1B). There is evidence to suggest that the relationship between age and cortical thickness is not a linear model, particularly for the anterior cingulate cortex (ACC). Indeed, ACC thickness is reduced until the age of 60 but is, thereafter, greater at older ages [55]. One hypothesis to explain the “reversal” of cortical thinning could be an increase in neuroplasticity [56,57]. When the brain is damaged, older adults can maintain cognitive and body functions by activating effective compensatory resources in response to this degradation [58]. Our results support this neuroplasticity as we observed in the aging cortex an enrichment of proteins playing an important role in neurogenesis, axonal guidance, dendrite outgrowth and synapse formation (e.g., Camk2d, Cacnb3, Dlgap1, Dpysl5, Gabra1, Gap-43) (Appendix A). We particularly showed that the expression of Gap-43, important for axonal growth cones and a marker of neuroplasticity [44,51], was higher in the aging cortex (Figure 5C). Interestingly, Gap-43 expression only increased at the protein level as its mRNA expression level remained stable in the aging cortex (Figure 5A,B), suggesting that this neuroplasticity could occur at the translational level rather than through transcriptional regulation. However, cortical thickness might not only be supported by neuroplasticity. Inflammation increases with age and could cause an age-dependent related thickness. Cortical thinning has been globally associated with level of systemic inflammation in older persons without dementia [59], but other studies have shown a positive relationship between inflammatory markers and cortical thickness, specifically for the ACC [54,55,60,61,62]. For instance, TNF-α, interleukine-8 and CCL2 correlate positively with ACC thickness [61]. Interestingly, we show that genes whose expression is increased at the RNA level in the cortex during aging are mainly involved in the immune response, including genes coding for inflammatory chemokines (Ccl8, Cxcl13) (Figure 2). We specifically showed that Ccl8 expression was significantly and specifically increased in the ACC during aging (Figure 3C,D). These results support that inflammation increases in the aging cortex and could potentially be implicated in the cortical thickening. It is important to note that this aging-dependent regulation occurs at the transcriptional level.

We performed RNA-seq and MS analyses on 6- and 24-month-old cortices. By decoupling mRNA and protein expression in the same cortex samples, we were able to identify distinct subsets of genes with different age-dependent regulations: i) genes regulated at the transcriptional level, with change in the mRNA level, and ii) genes regulated at a translational level, with stable mRNA expression during aging. Our RNA-seq analysis indicated that aging mainly triggers gene activation in the cortex, as 93% of the differentially expressed genes showed an increase in their mRNA level (Figure 2B,C). We also found that a significant number of these genes were related to sensory functions but also immune cell functions (Figure 2F). We were not able to confirm these transcriptomic changes at the proteomic level since only a minority of proteins encoded by DEGs were detected by MS (Appendix A). This matter was not due to post-mortem issues since the cortex samples were homogenized and equally fractionated at the same time for both transcriptomic and proteomic studies. In lieu, the immune cell population, being in the minority in the cerebral cortex, could logically explain the absence of the detection of these proteins by MS. We also showed that proteins detected by MS mainly originated from the most expressed transcripts, which did not include up-regulated genes related to the immune response (Appendix A; Appendix A). Moreover, transcriptomics and proteomics methods have very different levels of sensitivity and measurement, and it is still easier to detect mRNAs than their respective proteins. For instance, the GTEx Consortium performed a quantitative analysis of proteins from more than 12,000 genes across 32 normal human tissues [63]. They reported that many low-abundance transcripts were often under-detected at the protein level and that 16% of protein-coding genes with good RNA abundance (TPM > 32) were not detected at the protein level [63]. Immunostainings on brain slides were performed to override this limitation and confirm that genes up-regulated at the transcriptional level have, indeed, a higher level of proteins (Figure 3). It is interesting to note that genes whose expression was up-regulated during aging were statistically enriched in specific regions of chromosomes (Appendix A). This observation suggests that major epigenetic chromatin remodeling events occur in the cortex. It has been proposed that the loss of heterochromatin, which is characteristic of aging, can lead to changes in the genome architecture and, hence, increase the expression of genes residing in these regions [64]. It also might explain why we only identified 27 down-regulated genes in the aging cortex whose functions are mainly linked to neuron functions (Figure 2E,G). Together, our results support previously published transcriptomic data showing that functions affected in the human brain by the aging process were, for the up-regulated genes, the response to stress, glial cells and the immune response, and for the down-regulated genes, synapses, neurotransmission and the calcium signaling pathway [65,66,67].

Using MS, we also identified genes coding proteins that were differentially enriched in the aging cortex and exhibited stable mRNA levels during aging, suggesting that this specific subset of genes depends on aging-dependent translational regulation. We identified 208 proteins differentially enriched in the aging cortex (Figure 4A) and showed that 79% of these proteins (164/208) were involved in catabolic and metabolic processes such as the proteasome complex (Figure 4C). PSMD and PSMC proteins, which are proteasome 26S subunits involved in the degradation of misfolded and abnormal proteins [42,43], were specifically enriched in this subset (Appendix A). It has been reported that damaged and misfolded proteins tend to accumulate during the brain aging process and that the proteasome is responsible for the degradation of these proteins [68,69,70]. Age-related alterations in proteasome structure and activity contribute to brain aging by increasing the level of protein aggregation, potentially leading to the development of age-related neuropathologies such as Alzheimer’s disease with the aggregation of β-amyloid proteins. The enrichment of specific proteasome subunits in the aging cortex might modify proteasome biology, favor a decline of proteasome activity upon aging, and, hence, reduce protein degradation. We also showed that 21% of proteins (42/208) less enriched in the aging cortex, including ribosomal proteins, were mostly involved in translation and ribosome assembly (Figure 4D; Appendix A). During aging, the global translation rate decreases as the translation machinery becomes less efficient. This decrease is due to the deregulation of the stoichiometry of translation components that affect the abundance of ribosomes [10,11] and also the rate and accuracy of the translation [71,72]. Therefore, the age-dependent enrichment of ribosome components might affect protein homeostasis and promote a loss of proteostasis, one of the primary hallmarks of aging. Indeed, flawed translation has been described to lead to premature aging [73] and even to promote early symptoms of Alzheimer’s disease in aging mice [74].

By decoupling mRNA and protein expression, we have thus identified distinct subsets of genes that differentially adjust to cellular aging in the cerebral cortex. In the future, it will be important to determine for the first subset of genes the precise mechanisms of epigenetic and transcriptional regulation, and for the second subset of genes, the role of post-transcriptional, translational and degradation regulation in adjusting protein abundance. Single-cell methods, either at the RNA or protein level, have made significant advances over the last few years and could help to further discriminate which cell population (e.g., immune cells, glial cells or neurons) is targeted by these age-dependent mechanisms. As recently tackled [75], assessing the discordance between mRNA and protein abundance in neurodegenerative diseases may also reveal potential pathological protein drivers and foster new ways to investigate diseases that currently do not have any therapeutic options, such as Alzheimer’s disease. 

## Figures and Tables

**Figure 1 cells-12-00615-f001:**
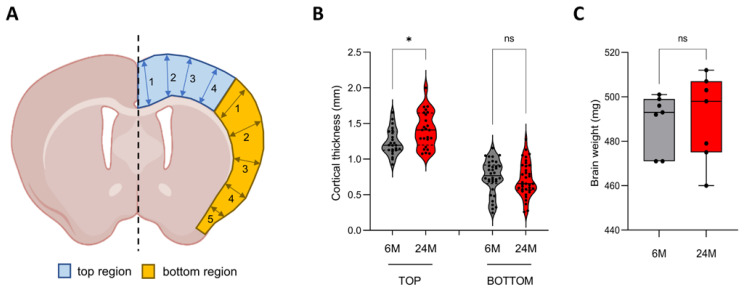
Anatomical characteristics of the aging brain. (**A**) Schematic depicting a coronal brain cut, with the two areas where cortical thickness was measured. The blue area represents the “top” of the cerebral cortex, comprising the anterior cingulate area, the motor area and the primary somatosensory area (limbs). The orange area represents the “bottom” of the cerebral cortex, with the supplemental somatosensory area (barrel and nose), the visceral area, the gustatory area and the piriform area. The thickness of the top and bottom areas was measured at different locations, as indicated by numbered arrows. (**B**) Coronal cuts of the whole brain of 6- and 24-month-old mice (respectively grey and red; n = 6 per age group; 3 males and 3 females) were stained with hematoxylin and eosin, and the thickness of the top and bottom areas was measured. (*) p < 0.0001 in one-way ANOVA; ns: not significant. (**C**) Graph representing the wet weight of the brain after dissection (n = 6 per age group; 3 males and 3 females). ns: not significant from unpaired t-test with equal means.

**Figure 2 cells-12-00615-f002:**
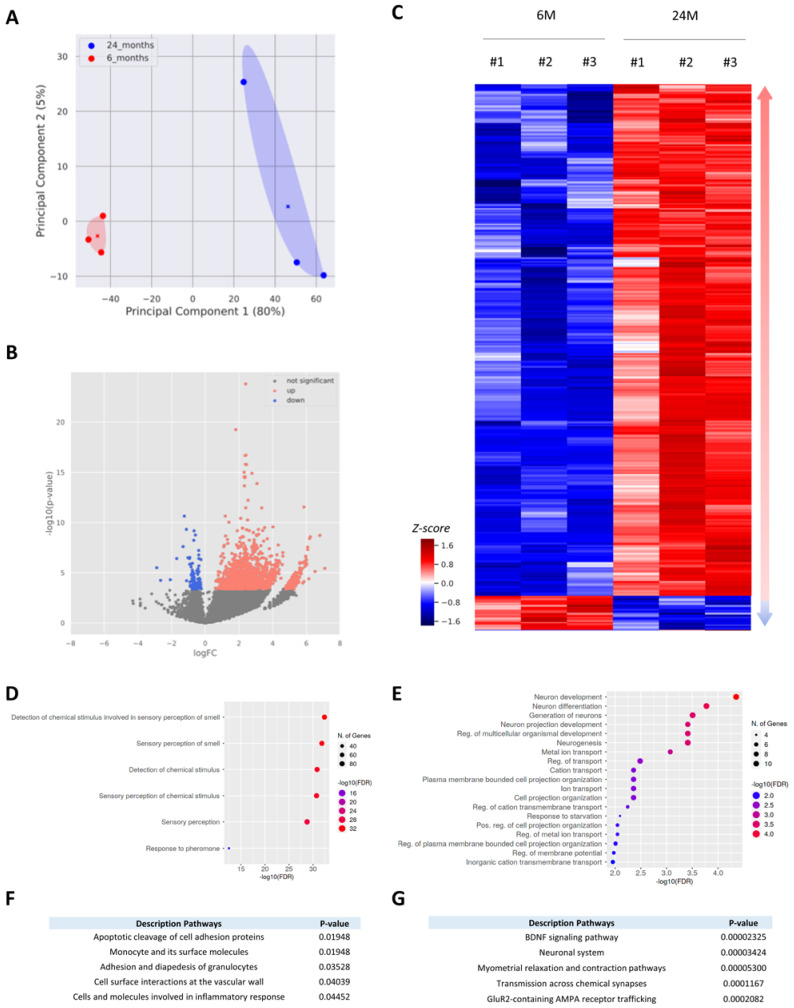
Age-related changes in the transcriptome signature of the cerebral cortex. (**A**) Principal component analysis (PCA, explaining 85% of the variance in total) for the gene expression of 6-month-old (red) and 24-month-old (blue) cortices (n = 3 per age group). The cross represents the centroid for each sample set. (**B**) Volcano plot of differentially expressed genes, mapping the 400 up-regulated genes (red) and 27 down-regulated genes (blue) at a 1% FDR. FC, fold change. (**C**) Hierarchical clustering of genes showing significant differential expression between 6- and 24-month-old cortices. Red and blue indicate relatively higher and lower expression, with genes independently scaled to a mean of zero. (**D**,**E**) Gene Ontology analysis of up-regulated (**D**) and down-regulated (**E**) genes using the ShinyGO tool (version 0.76.1). The pathway database used was the GO Biological Process. Parameters were set at a 0.05 FDR cut-off with no redundancy in GO terms. (**F**,**G**) Pathways enriched among up-regulated (**F**) and down-regulated (**G**) genes using the BioPlanet 2019 pathway database. Pathways are sorted by p-value ranking.

**Figure 3 cells-12-00615-f003:**
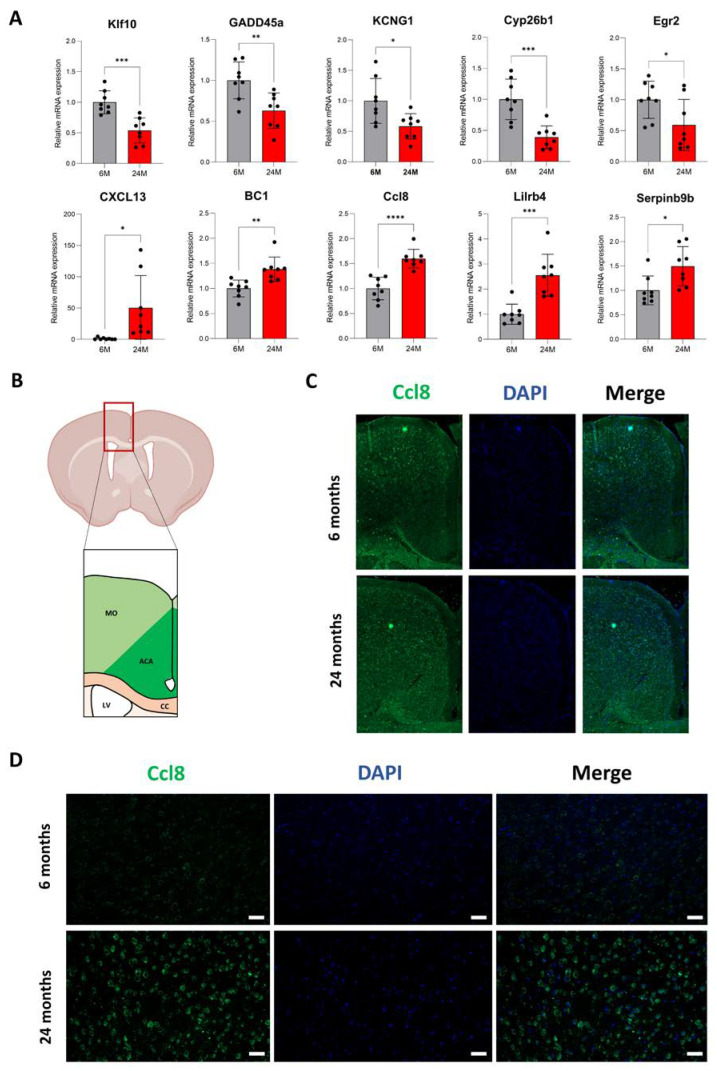
Validation of candidate genes differentially expressed in the aging cortex. (**A**) mRNA levels of Klf10, GADD45a, KCNG1, Cyp26b1, Egr2, CXCL13, BC1, Ccl8, Lilrb4b and Serpina9b were measured in cortices of 6-month-old (grey) and 24-month-old (red) mice by quantitative PCR and normalized to that of RPLP0 (n = 8 per age group; 4 males and 4 females). Results are indicated as mean ± SD. (*) p < 0.04, (**) p < 0.005, (***) p < 0.001, (****) p < 0.0001 from unpaired t-test with equal means. (**B**) Schematic representation of the cortex area analyzed by immunofluorescence. Motor area (MO), anterior cingulate area (ACA), corpus callosum (CC), lateral ventricle (LV). (**C**,**D**) Ccl8 immunofluorescence on coronal cuts of 6-month-old and 24-month-old brains. Images illustrate the representative staining in the whole top cortex area (**C**) and in the anterior cingulate area (**D**) with the Ccl8 antibody (green; left panel) and DAPI (blue; middle panel) (n = 3 per age group). The right panel constitutes a merged image of Ccl8 and DAPI signals. Scale bar: 50 μm.

**Figure 4 cells-12-00615-f004:**
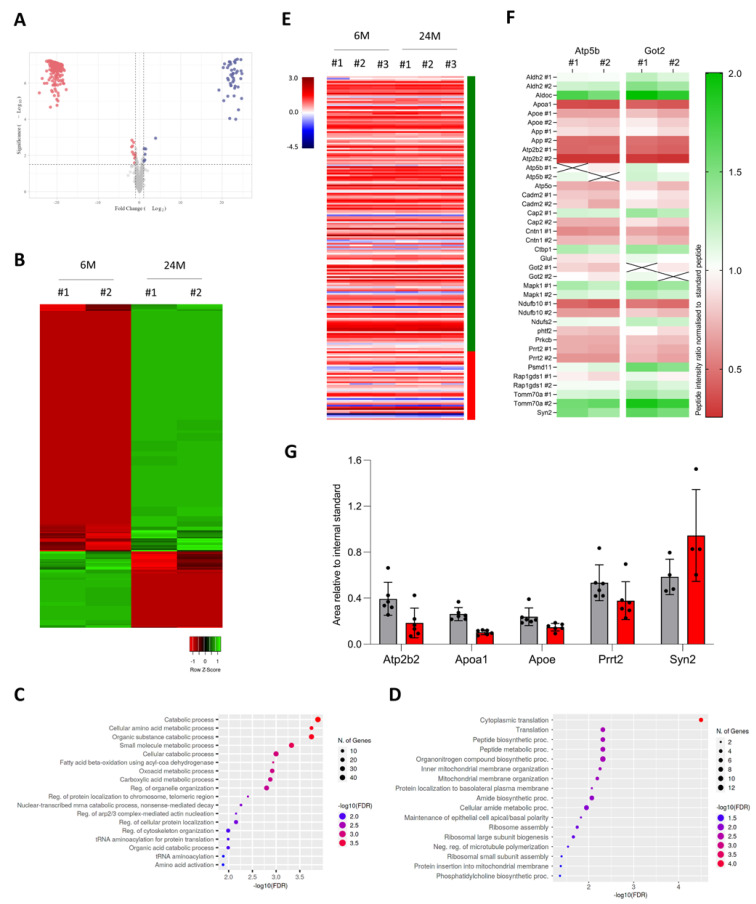
Age-related changes in the proteome of the cerebral cortex. (**A**) Volcano plot of the distribution of differentially enriched proteins, mapping the 164 down-regulated proteins (red) and 42 up-regulated proteins (blue). Data analysis was performed using Prostar software (log2 fold change threshold: 1, –log10 p-value: 1.3) (n = 2). (**B**) Hierarchical clustering of proteins showing significant differential enrichment in the aging cortex, based on the ratio of the area below the curve. Green and red indicate relatively high and low enrichment, with a ratio independently scaled to a mean of zero. (**C**,**D**) Gene Ontology analysis of proteins up-regulated (**C**) and down-regulated (**D**) in the aging cortex using the ShinyGO tool. The pathway database used was the GO Biological Process. Parameters were set at a 0.05 FDR cut-off with no redundancy in GO terms. (**E**) Heatmap of RNA transcript expression for each gene encoding differentially enriched proteins in 6 and 24-month-old cortices. Samples were independently scaled to a mean of zero. Green and red side bars indicate, respectively, up- and down-regulated proteins during cortex aging. (**F**) Heatmap of peptides quantified by PRM mass spectrometry. Each horizontal line represents one peptide. Green and red respectively indicate high and low expression. Values were based on the ratio of the area below the curve for 6- and 24-month-old cortex samples (n = 2 per age group) compared to that of the two Atp5b and Got2 control peptides. (**G**) Peptide quantification by PRM mass spectrometry. The area of each peptide was normalized by that of Got2 for 6-month-old (grey) and 24-month-old (red) cortex samples (n = 2 per age group; three technical replicates using two unique peptides). Results are indicated as mean ± SD.

**Figure 5 cells-12-00615-f005:**
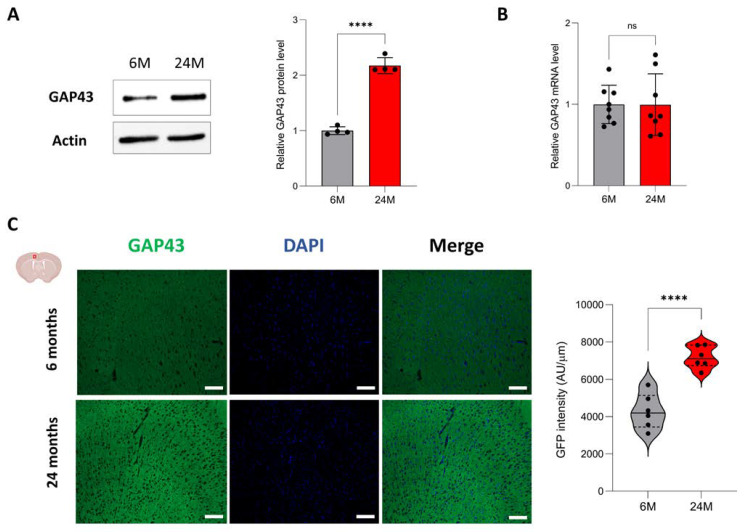
Analysis of GAP43 expression in the aging cortex. (**A**) Protein levels of GAP43 within the 6-month-old (grey) and 24-month-old (red) cortex samples were analyzed by Western blot (left panel). Actin was used as a loading control. On the right panel, quantification of the GAP43 signal (normalized by the actin signal) (n = 4 per age group). Results are indicated as mean ± SD. (****) p < 0.001 from unpaired t-test for equal means. (**B**) mRNA levels of GAP43 were measured by quantitative PCR and normalized to that of RPLP0 (n = 8 per age group). Results are indicated as mean ± SD. (**C**) GAP43 immunofluorescence on coronal cuts of 6-month-old and 24-month-old brains. Images illustrate the representative staining in the cortex motor area with the GAP-43 antibody (green; left panel) and DAPI (blue; middle panel). The right image panel constitutes a merged image of GAP43 and DAPI signals. A quantitative comparative analysis of GAP43 expression (n = 6 per age group) is displayed on the right. (****) p < 0.001 from unpaired t-test for equal means. Scale bar: 100 μm.

**Table 1 cells-12-00615-t001:** List of antibodies used for immunoblotting.

Antibody	Company	Cat n°	Dilution
Anti-Actin	Sigma Aldrich	A5441	1:20,000
Anti-GAP43	Abclonal	A19055	1:1000
Anti-mouse HRP	Biorad	170-6516	1:5000
Anti-rabbit HRP	Biorad	170-6515	1:5000

**Table 2 cells-12-00615-t002:** List of antibodies used for immunofluorescence.

Antibody	Company	Cat n°	Dilution
Anti-GAP43	Abclonal	A19055	1:50
Anti-CCL8	Abclonal	A6977	1:100
Alexa Fluor 488 anti-rabbit	Invitrogen	A-11017	1:1000

## Data Availability

The RNA-seq raw data have been deposited at the Gene Expression Omnibus (GEO) under the subseries entry GSE218637. The mass spectrometry proteomics data have been deposited to the ProteomeXchange Consortium via the PRIDE partner repository with the dataset identifier PXD038179.

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
