# Peer review of "Decoupling of mRNA and Protein Expression in Aging Brains Reveals the Age-Dependent Adaptation of Specific Gene Subsets"

_cells, 2023, doi:10.3390/cells12040615_

Round 1

Reviewer 1 Report

This study investigated the connection between changes of mRNA and protein expression in the brain by comparing the transcriptome and proteome of the mouse cortex during aging. It demonstrated that the increase of mRNA expression correlates with protein expression and the increase of cortical thickness during aging. Meanwhile, the characteristics of gene subgroups that have different regulatory effects on cell aging in the cerebral cortex were determined. The article is well organized, and its presentation is good, which deserves publication in Cells. However, some minor issues still need to be improved:

1.     There are writing mistakes in the manuscript: The numbers of CH3CN and NH4CO3 on pages 241 and 242 should be subscripts- CH3CN and NH4CO3; The acetonitrile writing of ACN appears on page 261, which should be unified; The writing method of µl includes both µL and µl, which should be unified too; The 1h30 on page 331?

2.     The third column of the table on page 341 is Reference, while the third column of the table on page 356 is Cat n°. Why are they named differently?

3.  Figure 2 D, E, and Figure 4 C, D are not clear enough, it is recommended to redraw them.

4.     Figure 3 and Figure 5 has a Merge image. It is suggested to indicate in the figure note.

Author Response

We thank the reviewer for her/his constructive criticisms and helpful suggestions. We have addressed each comment. The detailed point-by-point responses are provided below.

  1. There are writing mistakes in the manuscript: The numbers of CH3CN and NH4CO3 on pages 241 and 242 should be subscripts-CH3CN and NH4CO3; The acetonitrile writing of ACN appears on page 261, which should be unified; The writing method of µl includes both µL and µl, which should be unified too; The 1h30 on page 331?

We apologize for these writing mistakes. We corrected the chemical formula, revised the abbreviations, and unified the units in the overall Material & Methods section.

  1. The third column of the table on page 341 is Reference, while the third column of the table on page 356 is Cat n°. Why are they named differently?

We apologize for this issue. We unified the two tables and chose the catalog number as the title for the column.

  1. Figure 2 D, E, and Figure 4 C, D are not clear enough, it is recommended to redraw them.

We agree with the reviewer that the figures mentioned above are not of the highest resolution in the submitted pdf. Before the initial submission, we prepared all the figures in high quality resolution (tiff format; 1,000 dpi). However, the submission system does not allow us to upload this format and hence we had to submit a pdf version which had lower resolution. The high-resolution version of each figure will be incorporated in the article if the manuscript is accepted for publication. To ensure a better clarity, we also increased the size of the mentioned figures in the revised version of this manuscript.

  1. Figure 3 and Figure 5 has a Merge image. It is suggested to indicate in the figure note.

The legends of Figure 3 and Figure 5 now included the mention of the Merge image.

Reviewer 2 Report

This is a very interesting noble study with appropriate experiments. The interpretation is fine, with proper language and claims. Authors have identified distinct subsets of genes in aging mice brains, which will be more relevant to studying neurodegenerative diseases and finding therapeutic targets. The overall rigor and relevance to Cells is very strong.  I am supportive of this paper with minor modifications.

The introduction should be improved with the clinical aspects of older adults and neurodegenerative diseases. Aging is the greatest risk factor for cognitive decline in older adults can be included in the introduction.  Kumar P, Liu C, Hsu JW, Chacko S, Minard C, Jahoor F, Sekhar RV. Glycine and N-acetylcysteine (GlyNAC) supplementation in older adults improves glutathione deficiency, oxidative stress, mitochondrial dysfunction, inflammation, insulin resistance, endothelial dysfunction, genotoxicity, muscle strength, and cognition: Results of a pilot clinical trial. Clin Transl Med. 2021 Mar;11(3):e372.

Methods: The quantification of immunoblots should be mentioned in the methods.

Author Response

We thank the reviewer for her/his constructive criticisms and helpful suggestions. We have addressed each comment. The detailed point-by-point responses are provided below.

The introduction should be improved with the clinical aspects of older adults and neurodegenerative diseases. Aging is the greatest risk factor for cognitive decline in older adults can be included in the introduction.

As suggested by the reviewer, we included in the introduction a short section on aging, cognitive impairment, and neurodegenerative diseases. To avoid any confusion with pathological aging (neurodegenerative diseases), we also emphasized in the introduction that our study was performed during normal aging. Moreover, we included a more pathological aspect in the discussion of this manuscript.

The quantification of immunoblots should be mentioned in the methods.

We included a description of the immunoblot quantification in the Material & Methods section.